# Understanding Inter-Session Intentions via Complex Logical Reasoning

## Abstract

Understanding user intentions is crucial for enhancing product recommendations, navigation suggestions, and query reformulations. However, user intentions can be complex, involving multiple sessions and attribute requirements connected by logical operators such as And, Or, and Not. For example, a user may search for Nike or Adidas running shoes across various sessions, with a preference for the color purple. In another case, a user may have purchased a mattress in a previous session and is now seeking a corresponding bed frame without intending to buy another mattress. Prior research on session understanding has not sufficiently addressed how to make product or attribute recommendations for such complex intentions. In this paper, we introduce the task of logical session complex query answering, where sessions are treated as hyperedges of items, and we formulate the problem of complex intention understanding as a task of logical session complex queries answering (LS-CQA) on an aggregated hypergraph of sessions, items, and attributes. The proposed task is a special type of complex query answering task with sessions as ordered hyperedges. We also propose a new model, the Logical Session Graph Transformer (LSGT), which captures interactions among items across different sessions and their logical connections using a transformer structure. We analyze the expressiveness of LSGT and prove the permutation invariance of the inputs for the logical operators. We evaluate LSGT on three datasets and demonstrate that it achieves state-of-the-art results.

## 1 Introduction

Understanding user intention is a critical challenge in product search. During the product search process, a user's intention can be captured in many ways. Some intentions can be explicitly given through search keywords. For example, a user may use keywords like "Red Nike Shoes" to indicate the desired product type, brand, and color. However, search keywords may not always accurately reflect the user's intention, especially when they are unsure of what they want initially. To address this issue, session-based recommendation methods have been proposed to leverage user behavior information to make more accurate recommendations (Hidasi et al., 2015; Li et al., 2017).

User intentions are usually complex. On one hand, users often have explicit requirements for desired items, such as brand names, colors, sizes, and materials. For example, in Figure 1, query $q_1$ shows a user desiring Nike or Adidas products in the current session. On the other hand, users may spend multiple sessions before making a purchasing decision. For query $q_2$, the user spends two sessions searching for a desired product with an explicit requirement of purple color. Moreover, these requirements can involve logical structures. For instance, a user explicitly states that they do not want products similar to a previous session. In query $q_3$, a user has purchased a mattress in a previous session and is now looking for a wooden bed frame, without any intention of buying another mattress. With the help of logical operators like AND $\wedge$, OR $\vee$, and NOT $\neg$, we can describe the complex intentions by using a complex logical session query, like $q_1$, $q_2$, and $q_3$ in Figure 1.

Furthermore, there are scenarios we also want to know the product attributes based on sessions. For instance, in query $q_4$ in Figure 1, we want to identify the material types of the products desired in the current session. Similarly, in query $q_5$, given two sessions, we want to determine the brand names of the products desired in both sessions. To deal with them, we can also describe these queries by logic expressions with the help of variables. For example, we can use the variable $V_1$ to represent the products and $V_?$ to represent the attribute associated with the product $V_1$. Consequently, the attribute

| Complex Queries | Interpretations |
|---|---|
| $q_1 = V_? : Session(Item_1, Item_2, ..., Item_k, V_?)$ $\wedge (Brand(V_?, Nike) \vee Brand(V_?, Adidas))$ | Find the desired next item of a session with the brand Nike or Adidas. |
| $q_2 = V_? : Session(Item_{1,1}, Item_{1,2}, ..., Item_{1,m}, V_?) \wedge$ $Session(Item_{2,1}, Item_{2,2}, ..., Item_{2,n}, V_?) \wedge Color(V_?, Purple)$ | Find the desired next item of both $session_1$ and $session_2$ with purple color. |
| $q_3 = V_? : Session(Item_{1,1}, Item_{1,2}, ..., Item_{1,m}, V_?) \wedge$ $\neg Session(Item_{2,1}, Item_{2,2}, ..., Item_{2,n}, V_?) \wedge Material(V_?, Wood)$ | Find the item with wooden material that is desired by the $session_1$ but is not desired by $session_2$. |
| $q_4 = V_?, \exists V_1 : Session(Item_1, Item_2, ..., Item_k, V_1)$ $\wedge Material(V_1, V_?)$ | Find the material type of the product that a session desires. |
| $q_5 = V_?, \exists V_1 : Session(Item_{1,1}, Item_{1,2}, ..., Item_{1,m}, V_1) \wedge$ $Session(Item_{2,1}, Item_{2,2}, ..., Item_{2,n}, V_1) \wedge Brand(V_1, V_?)$ | Find the brand name the product that is desired by both the $session_1$ and by the $session_2$. |

Figure 1: Example complex queries involving multiple sessions and various product attributes.

recommendation task with complex user intention can be formulated as a multi-hop logical query answering task, where we inquire about what $V_?$ would be such that there exists a certain product $V_1$ with the product attribute $V_?$ that is desired in the given sessions.

To systematically answer queries with complex user intentions, we formally propose the task of logical session complex query answering (LS-CQA). This can be seen as an extension of the Complex Query Answering (CQA) problem to multi-relational hypergraph data, where sessions are treated as ordered hyperedges of items. The task of product or attribute recommendation under complex intention is reformulated as a task of answering logical queries on an aggregated hypergraph of sessions, items, and attributes. Figure 2 (C) provides an example of such an aggregated hypergraph, where each session is represented as a hyperedge connecting the corresponding items.

In addition to utilizing CQA methods with N-ary facts, such as NQE proposed by Luo et al. (2023), another more reasonable approach to LS-CQA is to employ a session encoder. Recent studies (Li et al., 2021; Huang et al., 2022; Zhang et al., 2023) ave shown the effectiveness of session encoders in encoding sessions and generating session representations. However, the neural session encoders tend to conduct implicit abstraction of products during the session encoding process (Zhang et al., 2023). The logical query encoder can only access the abstracted session representations, resulting in a lack of capturing the interactions between items in different sessions during the query encoding.

Motivated by this, we introduce the Logical Session Graph Transformer (LSGT) as an approach for encoding complex queries sessions as hypergraphs. Building upon the work by Kim et al. (2022), we transform items, sessions, relation features, session structures, and logical structures into tokens, which are then encoded using a standard transformer model. This transformation enables us to effectively capture interactions among items in different sessions through the any-to-any attention mechanisms in transformer models. By analyzing the Relational Weisfeiler-Lehman by Barceló et al. (2022); Huang et al. (2023), we provide theoretical justification for LSGT, demonstrating that it possesses the expressiveness of at least 1-RWL, and has at least same expressiveness as existing logical query encoders that employ message-passing mechanisms for logical query encoding in WL test. Meanwhile, LSGT maintains the property of operation-wise permutation invariance, similar to other logical query encoders. To evaluate LSGT, we have conducted experiments on three evaluation datasets: Amazon, Diginetica, and Dressipi. Experiment results show that the transformer-based query encoding models have similar performances. SQE with Transformer (Bai et al., 2023) and LSGT perform comparably on the EPFO queries, and they perform better than the graph-based query encoders with session encoders. While the linearization strategy of LSGT is better on the queries involving negations, achieving an improvement from 1.92 to 2.93 in MRR. We also discovered that the linearization strategy used in LSGT has better compositional generalization capability. In general, the contribution of this paper can be summarized as follows:

- We extend complex query answering (CQA) to hypergraphs with sessions as ordered hyperedges (LS-CQA) for describing and solving the product and attribute recommendations with complex user intentions. We also constructed three corresponding scaled datasets with the full support of first-order logical operators (intersection, union, negation) for evaluating CQA models on hypergraphs with ordered hyperedges and varied arity.

- We propose a new method, logical session graph transformer (LSGT). We use a new linearization strategy of hypergraph queries, which uses tokens and identifiers to uniformly represent the items, sessions, logical operators, and their relations, and then uses a standard transformer structure to encode them.

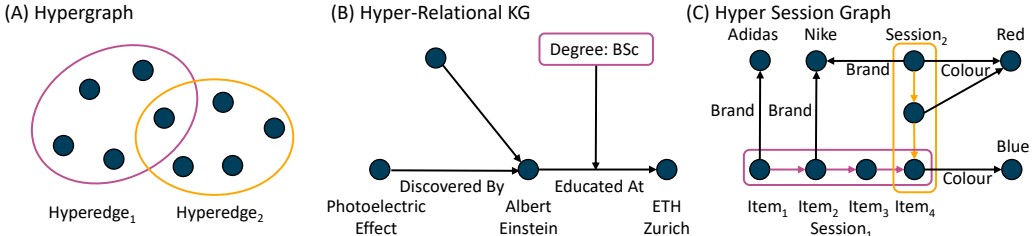

Figure 2: This figure shows the connections and differences between general hypergraphs, hyper-relational knowledge graphs, and the hyper-session graph in our problem.

- We conducted experiments on Amazon, Digintica, and Dressipi to show that existing Transformer-based models show similar results on 3 benchmarks despite different linearization strategies. Meanwhile, We also find the linearization of LSGT leads to an improvement in queries with negations and unseen query types. Meanwhile, We theoretically justify the expressiveness in the Weisfeiler-Lehman (WL) test and Relational Weisfeiler-Lehman (RWL) test, and we also prove the operator-wise permutation invariance of LSGT.

## 2 PROBLEM FORMULATION

### 2.1 LOGICAL SESSION COMPLEX QUERY ANSWERING

In previous work, complex query answering is usually conducted on a knowledge graph $\mathcal{G} = (\mathcal{V}, \mathcal{R})$. However, in our aggregated hypergraph, there are not only items but also sessions and attribute values. Because of this, the graph definition is $\mathcal{G} = (\mathcal{V}, \mathcal{R}, \mathcal{S})$. The $\mathcal{V}$ is the set of vertices $v$, and the $\mathcal{R}$ is the set of relation $r$. The $\mathcal{S}$ is the set of sessions that can be regarded as a set of directed hyperedges. To describe the relations in logical expressions, the relations are defined in functional forms. Each relation $r$ is defined as a function, and it has two arguments, which represent two items or attributes $v$ and $v'$. The value of function $r(v, v') = 1$ if and only if there is a relation between the items or attributes $v$ and $v'$. Each session $s \in \mathcal{S}$ is the sequence of vertices where $s(v_1, v_2, ..., v_n) = 1$ if and only if $v_1, v_2, ..., v_n$ appeared in the same session.

The queries are defined in the first-order logical (FOL) forms. In a first-order logical expression, there are logical operations such as existential quantifiers $\exists$, conjunctions $\wedge$, disjunctions $\vee$, and negations $\neg$. In such a logical query, there are anchor items or attribute $V_a \in \mathcal{V}$, existential quantified variables $V_1, V_2, ...V_k \in \mathcal{V}$, and a target variable $V_? \in \mathcal{V}$. The knowledge graph query is written to find the answer $V_? \in \mathcal{V}$, such that there exist $V_1, V_2, ...V_k \in \mathcal{V}$ satisfying the logical expression in the query. For each query, it can be converted to a disjunctive normal form, where the query is expressed as a disjunction of several conjunctive expressions:

$$q[V_?] = V_?.\exists V_1, ..., V_k : c_1 \vee c_2 \vee ... \vee c_n, \tag{1}$$

$$c_i = e_{i1} \wedge e_{i2} \wedge ... \wedge e_{im}. \tag{2}$$

Each $c_i$ represents a conjunctive expression of literals $e_{ij}$, and each $e_{ij}$ is an atomic or the negation of an atomic expression in any of the following forms: $e_{ij} = r(v_a, V)$, $e_{ij} = \neg r(v_a, V)$, $e_{ij} = r(V, V')$, or $e_{ij} = \neg r(V, V')$. The atomics $e_{ij}$ can be also hyper N-ary relations between vertices indicating that there exists a session among them. In this case, the $e_{ij} = s(v_1, v_2, ..., v_n, V)$ or its negations $e_{ij} = \neg s(v_1, v_2, ..., v_n, V)$. Here $v_a$ and $v_i \in V_a$ is one of the anchor nodes, and $V, V' \in \{V_1, V_2, ..., V_k, V_?\}$ are distinct variables satisfying $V \neq V'$. When a query is an existential positive first-order (EPFO) query, there are only conjunctions $\wedge$ and disjunctions $\vee$ in the expression (no negations $\neg$). When the query is a conjunctive query, there are only conjunctions $\wedge$ in the expressions (no disjunctions $\vee$ and negations $\neg$).

## 3 RELATED WORK

### 3.1 HYPER-RELATIONAL GRAPH REASONING

The reasoning over hyper-relational KG proposed by Alivanistos et al. (2022), they extend the multi-hop reasoning problem to hyper-relational KGs and propose a method, StarQE, to embed and answer

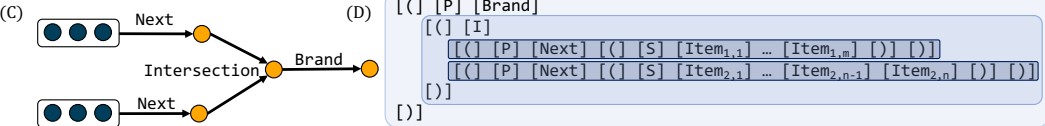

(A) $q_1 = V_?, \exists V_1 : Session(Item_{1,1}, \ldots, Item_{1,m}, V_1) \wedge Session(Item_{2,1}, \ldots, Item_{2,n}, V_1) \wedge Brand(V_1, V_?)$

(B) Find the brand name the product that is desired by both the $session_1$ and by the $session_2$.

(C)

(D)
```
[() [P] [Brand]
    [() [I]
        [() [P] [Next] [() [S] [Item_{1,1}] … [Item_{1,m}] ()] ()]
        [() [P] [Next] [() [S] [Item_{2,1}] … [Item_{2,n-1}] [Item_{2,n}] ()] ()]
    ()]
()]
```

Figure 3: The illustration of different query embedding methods. (A) The logical session complex query expressed in the first-order logic form. (B) The interpretations on the logical session complex query. (C) The computational graph of the complex query proposed by Hamilton et al. (2018); (D) The linearization of the computational graph to token proposed by Bai et al. (2022a)

hyper-relational conjunctive queries using Graph Neural Networks and query embedding techniques. The StarQE conducts message-passing over the quantifier of the hyper-relations in KG, which cannot be directly used for encoding the hyper-relations of the session. Luo et al. (2023) propose a novel Nary Query Embedding (NQE) model for complex query answering over hyper-relational knowledge graphs, which can handle more general n-ary FOL queries including existential quantifiers, conjunction, disjunction, and negation. The encoder design of NQE is more general to N-ary facts, thus it can be directly used for encoding the sessions as hyper-edges.

## 3.2 COMPLEX QUERY ANSWERING

Previous literature on logical query answering mainly focuses on knowledge graphs (Hamilton et al., 2018; Ren et al., 2020; Arakelyan et al., 2021). Various methods are proposed to deal with the incompleteness issue of knowledge graphs by using the existing facts in the KG to generalize to the facts that are not in the KG but highly likely to be true. Recent advances in query embedding (Hamilton et al., 2018; Ren et al., 2020; Ren & Leskovec, 2020) methods have shown promise in answering logical queries on large-scaled graph-structured data effectively and efficiently. However, they cannot be directly used for answering queries with hyperedges, or in other words N-ary facts. Meanwhile, there are also methods (Alivanistos et al., 2022; Luo et al., 2023) that can perform robust reasoning on hyper-relational knowledge graphs, which is illustrated in Figure 2 (B). Because of the fundamental differences between the hyper-relational knowledge graphs and the hypergraphs of sessions, not all of them can be directly adopted for this task. Recently, there is also new progress on query encoding that is orthogonal to this paper, which puts a focus on the neural encoders for complex queries. Xu et al. (2022) propose a neural-symbolic entangled method, ENeSy, for query encoding. Yang et al. (2022) propose to use Gamma Embeddings to encode complex logical queries. Liu et al. (2022) propose to use pre-training on the knowledge graph with kg-transformer and then conduct fine-tuning on the complex query answering. Meanwhile, query decomposition (Arakelyan et al., 2021) is another way to deal with the problem of complex query answering. In this method, the probabilities of atomic queries are first computed by a link predictor, and then continuous optimization or beam search is used to conduct inference time optimization. Moreover, Wang et al. (2023) propose an alternative to query encoding and query decomposition, in which they conduct message passing on the one-hop atomics to conduct complex query answering. Recently a novel neural search-based method QTO (Bai et al., 2022b) is proposed. QTO demonstrates impressive performance CQA. There are also neural-symbolic query encoding methods proposed (Xu et al., 2022; Zhu et al., 2022). In this line of research, their query encoders refer back to the training knowledge graph to obtain symbolic information from the graph. LogicRec (Tang et al., 2023) discusses the problem of recommending highly personalized items based on complex logical requirements, which current recommendation systems struggle to handle.

## 3.3 SESSION ENCODERS

In recent literature, various methods have been proposed to reflect user intentions and build better recommendation systems using session history. Because of the nature of sequence modeling, various methods utilize recurrent neural networks (RNNs) and convolutions neural networks (CNNs) to model session data (Hidasi et al., 2015; Li et al., 2017; Liu et al., 2018; Tang & Wang, 2018). Recent developments in session-based recommendation have focused on using Graph Neural Networks (GNNs) to extract relationships and better model transitions within sessions (Li et al., 2021; Guo et al., 2022; Huang et al., 2022). Wu et al. (2019) were the first to propose using GNNs to cap-

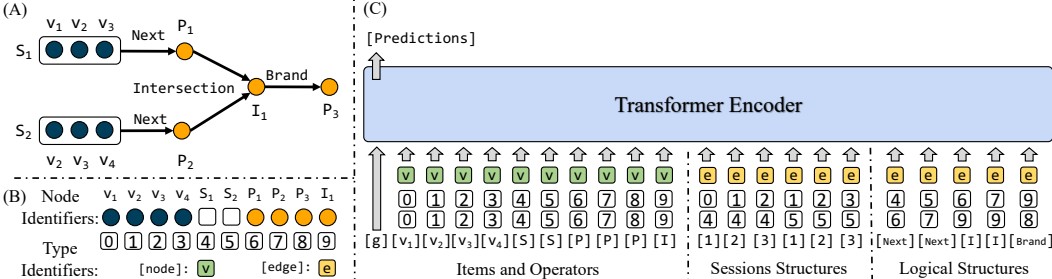

Figure 4: This figure shows the method of LSGT.

ture complex transitions with graph structures, and subsequent research has incorporated position and target information, global context, and highway networks to further improve performance (Pan et al., 2020; Xia et al., 2021). However, previous efforts have focused more on the message-passing part and less on designing effective readout operations to aggregate embeddings to the session-level embedding. According to Zhang et al. (2023), current readout operations have limited capacity in reasoning over sessions, and the performance improvement of GNN models is not significant enough to justify the time and memory consumption of sophisticated models. So Zhang et al. (2023) proposed a pure attention-based method Atten-Mixer to conduct session recommendations.

## 4 LOGICAL SESSION GRAPH TRANSFORMER

In this session, we describe the logical session graph transformer (LSGT) that is specialized for encoding logical queries involving sessions. In LSGT, the node and edge features, session structures, and logical structures are all converted into tokens and identifiers. Subsequently, these tokens and identifiers serve as input to a standard transformer encoder model.

### 4.1 ITEMS, SESSIONS, AND OPERATORS TOKENS

The first step in LSGT involves assigning node identifiers to every item, session, and operator. For instance, in Figure 4, there are two sessions, $S_1$ and $S_2$, with items $[v_1, v_2, v_3]$ and $[v_2, v_3, v_4]$, respectively. The computational graph then uses relational projection operators $P_1$ and $P_2$ to find the two sets of next items desired by $S_1$ and $S_2$, respectively. Once all items, sessions, and operators have been identified, each is assigned a unique node identifier. For example, $v_1$ to $v_4$ are assigned identifiers from 0 to 3, $S_1$ and $S_2$ are assigned identifiers 4 and 5, projections from $P_1$ to $P_3$ are assigned identifiers from 6 to 8, and intersection operation $I_1$ is assigned identifier 9.

In general, when there are $n$ nodes denoted by their identifiers as $\{p_0, p_1, ..., p_n\}$, their node features are assigned as follows: if $p_i$ is an item, its features are assigned to its item embedding. If $p_i$ is a session $S_j$, it is assigned an embedding of $[S]$. If $p_i$ is a neural operator, it is assigned the operator embedding from $[P]$, $[I]$, $[N]$, or $[U]$ based on its operation type. The feature matrix for these $n$ nodes is then denoted as $\boldsymbol{X}_p \in \mathbb{R}^{n \times d_1}$. Additionally, each node identifier is associated with random orthonormal vectors (Kim et al., 2022), denoted as $\boldsymbol{P}_p \in \mathbb{R}^{n \times d_2}$. All nodes are assigned the type identifier of $[node]$, which means that they are the nodes in the computational graph. The token type embedding for vertices is denoted as $T_{[node]} \in \mathbb{R}^{d_3}$. The input vectors for the transformer are concatenations of node features, the random orthonormal vectors, and token type embeddings, where node identifiers vectors are repeated twice: $\boldsymbol{X}_u^v = [\boldsymbol{X}_p, \boldsymbol{P}_p, \boldsymbol{P}_p, \boldsymbol{T}_{[node]}] \in \mathbb{R}^{n \times (d_1 + 2d_2 + d_3)}$.

### 4.2 SESSION STRUCTURE TOKENS

In this part, we describe the process of constructing the input tokens to indicate the session structure, namely which items are in which session in which position. Suppose the item $p$ is from session $q$ and at the position of $r$, and there are $m$ item-session correspondences in total. First, we use positional encoding $Pos(\mathrm{r}) \in \mathbb{R}^{d_1}$ to describe the positional information. Meanwhile, as the item and sessions are associated with their node identifiers $p$ and $q$, we use the node identifier vectors $P_p \in \mathbb{R}^{d_2}$ and $P_q \in \mathbb{R}^{d_2}$ to represent them. Meanwhile, this token represents a correspondence between two nodes, so we use the $[edge]$ token type embedding to describe this $T_{[edge]} \in \mathbb{R}^{d_3}$. As there are in total $m$

Table 1: The detailed statistics of the constructed hypergraph on sessions, items, and their attribute values are shown in the table.

| Dataset | # Edges | # Vertices | # Sessions | # Items | # Attributes | # Relations |
|---------|---------|-----------|-----------|---------|-------------|------------|
| Amazon | 8,004,984 | 2,431,747 | 720,816 | 431,036 | 1,279,895 | 10 |
| Diginetica | 1,387,861 | 266,897 | 12,047 | 134,904 | 125,204 | 3 |
| Dressipi | 2,698,692 | 674,853 | 668,650 | 23,618 | 903 | 74 |

of item-session correspondences, we concatenate them together to obtain the input vectors for the tokens representing session structures: $\boldsymbol{X}^s_{(p,q,r)} = [Pos(\mathrm{r}), \boldsymbol{P}_p, \boldsymbol{P}_q, \boldsymbol{T}^{[edge]}] \in \mathbb{R}^{m \times (d_1 + 2d_2 + d_3)}$.

### 4.3 LOGICAL STRUCTURE TOKENS

In this part, we describe the process of constructing the input for tokens to indicate the logical structures. As shown in Figure 4, in an edge representing a logical operation, there are two nodes $p$ and $q$ respectively. If the logical operation is projection, then the edge feature is assigned with relation embedding [Rel]. Otherwise, the edge feature is assigned with the operation embedding from [P], [I], [N], and [U] accordingly. The edge feature is denoted as $R_r \in \mathbb{R}^{d_1}$. Similarly, we use the node identifier vectors $P_p \in \mathbb{R}^{d_2}$ and $P_q \in \mathbb{R}^{d_2}$ to represent involved nodes $p$ and $q$. Meanwhile, this token represents an edge in the computational graph, so we also associate it with token type embedding $T_{[edge]} \in \mathbb{R}^{d_3}$ to describe it. Suppose there are in total $w$ such logical edges, we concatenate them together to obtain the input vectors for the tokens representing logical structure: $\boldsymbol{X}^l_{(p,q,r)} = [\boldsymbol{R}_r, \boldsymbol{P}_p, \boldsymbol{P}_q, \boldsymbol{T}_{[edge]}] \in \mathbb{R}^{w \times (d_1 + 2d_2 + d_3)}$.

### 4.4 TRAINING LSGT

After obtaining the three parts describing the items, session structures, and logical structures, we concatenate them together $\boldsymbol{X} = [X_{[graph]}, \boldsymbol{X}^v, \boldsymbol{X}^s, \boldsymbol{X}^l] \in R^{(m+n+w+1) \times (d_1 + 2d_2 + d_3)}$, and use this matrix as the input for a standard transformer encoder for compute the query encoding of this complex logical session query. Then we append a special token [graph] with embedding $X_{[graph]} \in \mathbb{R}^{d_1 + 2d_2 + d_3}$ at the beginning of the transformer and use the token output of the [graph] token as the embedding of the complex logical session query. To train the LSGT model, we compute the normalized probability of the vertice $a$ being the correct answer of query $q$ by using the softmax function on all similarity scores,

$$p(q, a) = \frac{e^{<e_q, e_a>}}{\sum_{a' \in V} e^{<e_q, e_{a'}>}}. \tag{3}$$

Then we construct a cross-entropy loss to maximize the log probabilities of all correct pairs:

$$L = -\frac{1}{N} \sum_i \log p(q^{(i)}, a^{(i)}). \tag{4}$$

Each $(q^{(i)}, a^{(i)})$ denotes one of the positive query-answer pairs, and there are $N$ pairs.

### 4.5 THEORETICAL PROPERTIES OF LSGT

In this part, we analyze the theoretical properties of LSGT, focusing on two perspectives. First, we analyze the expressiveness of LSGT compared to baseline methods in Theorem 1 and 2. Second, we analyze whether LSGT has operator-wise permutation invariant, and this property is important in query encoding as operators like Intersection and Union are permutation invariant to inputs in Theorem 3. We prove the following theorems in the Appendix A of LSGT:

**Theorem 1.** *When without considering the relation types in the query graph, the expressiveness of the LSGT encoder is at least the same as that of the encoder that combines a session encoder followed by a logical query encoder under Weisfeiler-Lehman tests (Maron et al., 2019).*

**Theorem 2.** *When considering the query graphs are multi-relational graphs with edge relation types, the expressiveness of the LSGT encoder is also at least as powerful as 1-RWL, namely the expressiveness of R-GCN and CompGCN. (Barceló et al., 2022; Huang et al., 2023).*

**Theorem 3.** *LSGT can approximate a logical query encoding model that is operator-wise input permutation invariant.*

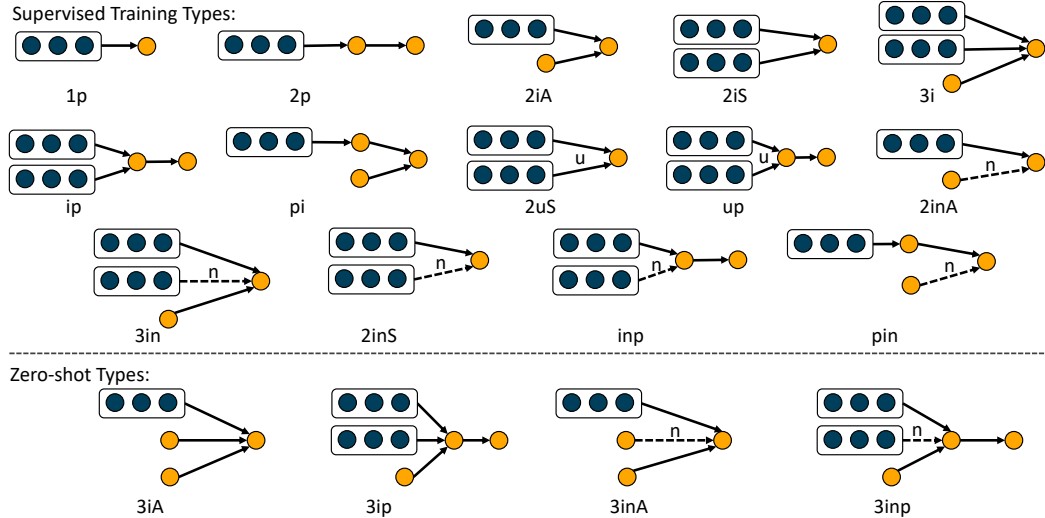

Figure 5: The query structures are used for training and evaluation. For brevity, the $p$, $i$, $n$, and $u$ represent the projection, intersection, negation, and union operations. The query types are trained and evaluated under supervised settings.

## 5 EXPERIMENT

We use three public datasets from KDD-Cup [1] (Jin et al., 2023), Diginetica [2], and Dressipi [3] for evaluation. The number of items, sessions, and relations are reported in Table 1. Following previous work Ren & Leskovec (2020); Wang et al. (2021); Bai et al. (2023), we use eighty percent of the edges for training, ten percent of edges for validation, and the rest of the edges as testing edges. As shown in Figure 5, we conduct sampling of fourteen types of logical session queries by using the sampling algorithm described by (Bai et al., 2023). The number of queries is shown in Table 5. Each of the queries has a concrete meaning. For example, the 1p queries are vanilla session-based product recommendations, and the 2p queries aim to recommend product attributes based on a single session history. A detailed explanation of the query types is shown in Appendix B.

### 5.1 BASELINE MODELS

We briefly introduce the baseline query encoding models that use various neural networks to encode the query into embedding structures. Here are the baseline models for the complex query-answering models: (1) NQE Luo et al. (2023) is a method that can be used to encode N-ary facts from the KG; (2) SQE (Bai et al., 2023) use sequence encoder to encode linearized complex queries.

In the hyper-relational session-product-attribute graph, each session can be formulated as a directed hyper-relation among various entities. Because of this, we construct the relation of NEXT connecting the items that are browsed in the session following the corresponding order. We employed a state-of-the-art session encoder to model the item history within a session. The session encoder takes into account the temporal dependencies and context of products, effectively creating a contextual representation of the entire session: (1) Q2P (Bai et al., 2022a) uses multiple vectors to encode the queries; (2) FuzzQE (Chen et al., 2022) use fuzzy logic to represent logical operators. Meanwhile, the previous query encoder cannot be directly used for encoding session history as hyper-relations, so we incorporate them with session encoders. For the session encoders, we leverage sequence-based encoder GRURec (Tan et al., 2016), GNN-based session encoder SR-GNN (Wu et al., 2019), and the state-of-the-art attention-based session encoder Attention-Mixer (Zhang et al., 2023).

---

[1] https://www.aicrowd.com/challenges/amazon-kdd-cup-23-multilingual-recommendation-challenge
[2] https://competitions.codalab.org/competitions/11161
[3] https://dressipi.com/downloads/recsys-datasets

Table 2: The performance in the mean reciprocal ranking of LSGT compared with the baseline models of SQE and NQE with different backbone modules.

| Dataset | Query Encoder | Session Encoder | 1p | 2p | 2ia | 2is | 3i | pi | ip | 2u | up | Ave. EPFO |
|---|---|---|---|---|---|---|---|---|---|---|---|---|
| Amazon | FuzzQE | GRURec | 11.99 | 6.96 | 54.79 | 88.47 | 68.13 | 16.73 | 14.49 | 10.02 | 6.87 | 30.94 |
| | | SRGNN | 13.52 | 7.93 | 54.12 | 85.45 | 67.56 | 18.62 | **20.46** | 10.82 | 7.24 | 31.75 |
| | | Attn-Mixer | 16.79 | 7.96 | 55.76 | **89.64** | 69.16 | 14.87 | 9.93 | 13.83 | 7.20 | 31.68 |
| | Q2P | GRURec | 11.60 | 6.83 | 34.73 | 67.64 | 42.18 | 16.66 | 13.82 | 8.54 | 5.82 | 23.09 |
| | | SRGNN | 13.94 | 7.69 | 35.89 | 69.90 | 44.61 | 16.19 | 16.44 | 10.20 | 6.46 | 24.59 |
| | | Attn-Mixer | 15.93 | 8.53 | 46.67 | 68.62 | 61.13 | 16.95 | 15.78 | 12.43 | 7.41 | 28.16 |
| | NQE | - | 5.60 | 2.50 | 48.40 | 77.98 | 63.06 | 2.20 | 1.80 | 4.20 | 3.00 | 23.19 |
| | SQE-Transformer | - | 16.09 | 8.30 | 53.90 | 72.26 | 64.48 | 17.54 | 16.80 | 13.86 | 7.35 | 30.07 |
| | SQE-LSTM | - | 16.59 | 7.45 | 55.60 | 86.81 | 69.11 | 17.86 | 19.04 | 13.46 | 6.87 | 32.53 |
| | LSGT (Ours) | | **17.73** | **9.10** | **56.73** | 84.62 | **69.39** | **19.39** | 19.47 | **15.40** | 7.86 | **33.26 (+0.73)** |
| Diginetica | FuzzQE | GRURec | 24.10 | 12.29 | 82.48 | 89.19 | 86.26 | 11.64 | 23.34 | 18.19 | 11.18 | 39.85 |
| | | SRGNN | 22.53 | 12.33 | 83.19 | 88.35 | 86.26 | 12.55 | 29.56 | 19.76 | 11.48 | 40.67 |
| | | Attn-Mixer | **33.87** | 11.89 | 82.94 | 88.94 | **86.36** | 12.28 | 28.21 | 24.78 | 10.81 | 42.23 |
| | Q2P | GRURec | 26.02 | **23.73** | 62.46 | 83.95 | 76.25 | 21.77 | 32.04 | 17.00 | 21.62 | 40.54 |
| | | SRGNN | 18.76 | 22.29 | 52.94 | 84.67 | 58.72 | 21.93 | 30.34 | 13.04 | **20.86** | 35.95 |
| | | Attn-Mixer | 34.87 | 24.36 | 55.00 | 87.09 | 58.46 | **22.81** | 31.26 | 25.76 | 21.60 | 40.13 |
| | NQE | - | 15.82 | 11.24 | 76.79 | 87.16 | 79.52 | 11.07 | 30.76 | 11.12 | 10.14 | 37.07 |
| | SQE-Transformer | - | 30.60 | 14.93 | 83.72 | **90.87** | 80.58 | 15.18 | 32.72 | 25.61 | 13.98 | 43.13 |
| | SQE-LSTM | - | 31.50 | 14.10 | 83.67 | 86.70 | 84.76 | 14.46 | 30.08 | 21.92 | 12.53 | 42.19 |
| | LSGT (Ours) | | 32.00 | 15.27 | **83.34** | 90.61 | 86.05 | 15.62 | **33.80** | **26.34** | 14.45 | **44.16 (+1.03)** |
| Dressipi | FuzzQE | GRURec | 27.62 | 94.28 | 56.15 | 77.21 | 75.40 | 94.81 | 98.43 | 23.46 | 95.52 | 71.43 |
| | | SRGNN | 30.18 | 94.90 | 52.41 | 74.63 | 73.38 | 95.37 | 98.32 | 25.09 | 95.69 | 71.11 |
| | | Attn-Mixer | 30.60 | 94.80 | 57.17 | 78.14 | 75.94 | 94.83 | **98.57** | 24.39 | 95.69 | 72.24 |
| | Q2P | GRURec | **35.93** | 95.20 | 45.22 | 66.62 | 51.20 | 96.27 | 92.58 | 25.46 | 95.45 | 67.10 |
| | | SRGNN | 35.48 | 95.95 | 46.05 | 64.01 | 52.58 | 95.75 | 92.81 | 25.28 | 95.68 | 67.07 |
| | | Attn-Mixer | 37.92 | 96.04 | 47.06 | 66.47 | 50.91 | 96.22 | 94.88 | 26.16 | 95.75 | 67.93 |
| | NQE | - | 11.52 | 95.62 | 21.19 | 52.79 | 48.28 | 96.08 | 98.04 | 13.39 | 95.80 | 59.19 |
| | SQE-Transformer | - | 27.01 | 95.37 | 62.38 | **80.55** | **79.72** | 96.02 | 97.99 | 24.55 | 95.95 | 73.28 |
| | SQE-LSTM | - | 25.84 | 94.81 | 62.23 | 64.19 | 70.43 | 95.39 | 96.91 | 25.23 | 95.62 | 70.07 |
| | LSGT (Ours) | | 31.12 | **96.16** | **64.26** | 76.85 | 78.66 | **98.02** | 96.98 | **28.83** | **96.04** | **74.10 (+0.82)** |

## 5.2 EVALUATION

To precisely describe the metrics, we use the $q$ to represent a testing query and $\mathcal{G}_{val}$, $\mathcal{G}_{test}$ to represent the validation and the testing knowledge graph. Here we use $[q]_{val}$ and $[q]_{test}$ to represent the answers of query $q$ on the validation graph $\mathcal{G}_{val}$ and testing graph $\mathcal{G}_{test}$ respectively. Equation 5 describes how to compute the `Inference` metrics. When the evaluation metric is mean reciprocal ranking (MRR), then the $m(r)$ is defined as $m(r) = \frac{1}{r}$.

$$\texttt{Inference}(q) = \frac{\sum_{v \in [q]_{test}/[q]_{val}} m(\texttt{rank}(v))}{|[q]_{test}/[q]_{val}|}. \tag{5}$$

## 5.3 EXPERIMENT DETAILS

We maintain a consistent hidden size of $384$ for all models. This hidden size also corresponds to the size of session representation from session encoders in the baselines, as well as the query embedding size for the entire logical session query. We use the AdamW to train the models with a batch size of $512$. The models are optimized with a learning rate of $0.001$, except for the models with transformer structures, namely NQE, SQE-Transformer, and LSGT. These models are trained with a learning rate of $0.0001$ with a warm-up of 10000 steps. The SQE and LSGT models employ two layers of encoders. All models can be trained and evaluated on GPU with 24GB memory.

## 5.4 EXPERIMENT RESULTS

Table 2 compares the performance of different models with various backbones and configurations. Based on the experimental results, we can draw the following conclusions.

First, we observed that the proposed LSGT method outperforms all other models and is the current state-of-the-art for the task. Compared to models that utilize only session encoders followed by query encoders, LSGT can leverage item information across different sessions, which is crucial for achieving superior performance. Additionally, LSGT is better equipped to encode graph structural inductive bias due to its operation-wise permutation invariance property, thus it can perform better

Table 3: The performance in the mean reciprocal ranking of LSGT compared with the baseline models of SQE and NQE with different backbone modules on the queries involving negations.

| Dataset | Query Encoder | Session Encoder | 2ina | 2ins | 3in | inp | pin | Ave. Negative |
|---|---|---|---|---|---|---|---|---|
| Amazon | FuzzQE | GRURec | 10.11 | 10.39 | 50.83 | 30.11 | 3.72 | 21.03 |
| | | SRGNN | 12.02 | 11.08 | 51.37 | 30.79 | 6.06 | 22.26 |
| | | Attn-Mixer | 17.28 | 17.47 | 53.77 | 31.96 | 4.55 | 25.00 |
| | Q2P | GRURec | 9.56 | 10.21 | 18.59 | 30.83 | 3.87 | 14.61 |
| | | SRGNN | 11.57 | 11.97 | 20.08 | 35.07 | 4.42 | 16.62 |
| | | Attn-Mixer | 18.75 | 20.68 | 51.52 | **37.04** | 6.78 | 26.95 |
| | NQE | - | 5.00 | 5.10 | 48.16 | 30.26 | 2.10 | 18.12 |
| | SQE-Transformer | - | 18.15 | 18.88 | 55.83 | 34.76 | 8.21 | 27.16 |
| | SQE-LSTM | - | 18.42 | 19.10 | 56.99 | 33.67 | 7.45 | 27.13 |
| | LSGT (Ours) | | **20.98** | **22.00** | **60.70** | 35.95 | **8.84** | **29.69 (+2.93)** |
| Diginetica | FuzzQE | GRURec | 16.15 | 9.09 | 81.65 | 14.07 | 10.69 | 26.33 |
| | | SRGNN | 16.62 | 15.77 | 82.30 | 14.92 | 10.69 | 28.06 |
| | | Attn-Mixer | 22.49 | 23.99 | 82.33 | 13.87 | 9.17 | 30.37 |
| | Q2P | GRURec | 11.42 | 9.92 | 34.33 | 10.94 | 15.58 | 16.44 |
| | | SRGNN | 9.17 | 8.90 | 26.28 | 11.01 | 14.84 | 14.04 |
| | | Attn-Mixer | 19.44 | 23.84 | 26.72 | 11.05 | 15.12 | 19.23 |
| | NQE | - | 9.71 | 11.05 | 73.10 | 11.76 | 8.60 | 22.84 |
| | SQE-Transformer | - | 23.81 | 25.07 | 77.64 | 18.97 | 14.57 | 32.01 |
| | SQE-LSTM | - | 23.05 | 18.56 | 81.22 | 16.77 | 13.68 | 30.66 |
| | LSGT (Ours) | | **24.15** | **28.69** | **83.04** | **19.21** | **15.62** | **34.14 (+2.13)** |
| Dressipi | FuzzQE | GRURec | 20.73 | 20.97 | 50.50 | 97.37 | 92.69 | 56.45 |
| | | SRGNN | 23.50 | 23.68 | 50.47 | 97.36 | 92.89 | 57.58 |
| | | Attn-Mixer | 22.70 | 21.75 | 51.81 | 97.20 | 93.69 | 57.43 |
| | Q2P | GRURec | 20.75 | 25.64 | 24.75 | 97.97 | 63.86 | 46.59 |
| | | SRGNN | 20.04 | 24.35 | 26.11 | 97.70 | 64.04 | 46.45 |
| | | Attn-Mixer | **26.74** | **37.09** | 49.58 | **97.98** | 95.22 | 61.32 |
| | NQE | - | 8.58 | 10.60 | 14.49 | 97.40 | 94.56 | 45.13 |
| | SQE-Transformer | - | 21.15 | 25.08 | 63.23 | 97.59 | 95.41 | 60.49 |
| | SQE-LSTM | - | 21.03 | 24.76 | 63.14 | 97.73 | 94.50 | 60.23 |
| | LSGT (Ours) | | 25.58 | 30.66 | **65.93** | 97.74 | **96.30** | **63.24 (+1.92)** |

than SQE models. Meanwhile, LSGT demonstrates greater capability in handling queries involving negations than baseline models. It achieves more significant improvements on negation queries than EPFO queries, outperforming the best baseline model. Moreover, we observe that neural models can produce more accurate results when presented with additional information or constraints. For example, if we know that two sessions are expecting the same item, we can provide better product recommendations based on these two sessions rather than using a single session. This observation highlights the importance of modeling complex user intentions and the potential for improving service quality in real-world usage scenarios.

We further conduct ablation studies and evaluations on compositional generalization, and the results are shown in Table 6 and 7 respectively. Experiments show that both the logical structures and item orders are important for the task of LS-CQA, and the LSGT is effective in encoding both information. Meanwhile, LSGT also demonstrates strong compositional generalization capability and exceeds other transformer-based methods by 1.28 to 3.22 in MRR on three datasets.

## 6 CONCLUSION

In this paper, we presented a framework that models user intent as a complex logical query over a hyper-relational graph that describes sessions, products, and their attributes. Our framework formulates the session understanding problem as a logical session complex query on this graph and trains complex query-answering models to make recommendations based on the logical queries. We also introduced a novel method of logical session graph transformer (LSGT) and demonstrated its expressiveness and operator-wise permutation invariance. Our evaluation of fourteen intersection logical reasoning tasks showed that our proposed framework achieves better results on unseen queries and queries involving negations. Overall, our framework provides a flexible and effective approach for modeling user intent and making recommendations in e-commerce scenarios. Future work could extend our approach to other domains and incorporate additional sources of information to further improve recommendation accuracy.

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

# A    PROOFS

We give the proofs of **Theorem** 1, 2, and **Theorem** 3. Before proving these two theorems, we define the proxy graph of the computational graph we used in this paper that involves N-ary facts.

**Defintion (Proxy Graph)**    For each computational graph utilized by the query encoder, we can uniquely identify the corresponding proxy graph. This graph comprises binary edges without hyper-edges and consists of vertices representing items, sessions, and operators. The edges in the proxy graph can be categorized into three types: session edges, which connect item vertices to session vertices and utilize their position, such as $1, 2, \ldots, k$ as edge types; relational projection edges, which connect two vertices and employ the relation type as the edge type; and logical edges, which utilize the corresponding logical operation type as the edge type. It is important to note that the proxy graph is distinct for different computational graphs with N-ary facts.

**Defintion (Non-relational Argumented Proxy Graph)**    For each proxy graph, we create another graph called a Non-relational Argumented Proxy Graph. This graph includes all vertices in the original proxy. Meanwhile, the argument graph an additional node for each edge in the original graph, and it takes relation type as a node feature.

**Lemma 1.** *Encoding the complex session query by following the computational graph using a session encoder followed by query encoding is equivalent to performing message passing on the corresponding proxy graph.*

*Proof.* To prove this, we must analyze each operation in the original N-ary computational graph. For the session encoder part, the session representation is computed from the items it contains, which is equivalent to a message passing on the proxy graph with a unique aggregation function, namely the session encoder. For the intersection and union operations, the computational graph utilizes various specially designed logical operations to encode them, and they can be considered as messages passing over the proxy graph. Similarly, for the relational projection, the tail node aggregates information from the head node and relation type, which is also a message-passing process on the proxy graph. □

**Lemma 2.** *The encoding process of LSGT is equivalent to using TokenGT to encode the proxy graph.*

*Proof.* The encoding process of LSGT consists of three parts. First, the node tokens are used to identify and represent the items, sessions, and operators. Secondly, the logical structure tokens are employed to represent the logical connections between items and sessions. Finally, LSGT utilizes positional embedding as the token feature to describe the positional information of an item in a session. This process is equivalent to building an edge between the item and session and assigning its edge feature as the corresponding position embedding, which is done in the proxy graph. Therefore, encoding logical session graphs using LSGT is equivalent to using TokenGT on the proxy graph. □

**Lemma 3.** *Suppose the $G_1$ and $G_2$ are two proxy graphs, and $G_1'$ and $G_2'$ are two non-relational argument proxy graphs converted from $G_1$ and $G_2$ respectively. Then $G_1 = G_2 \longleftrightarrow G_1' = G_2'$.*

*Proof.* The direction $G_1 = G_2 \to G_1' = G_2'$ is trivial because according to the definition, the conversion process is deterministic. We focus on the reverse side: $G_1 = G_2 \leftarrow G_1' = G_2'$. We try to prove it by contradiction. Suppose $G_1 \neq G_2$ but $G_1' = G_2'$. Without losing generality, we can suppose there is an edge $(u, v, r) \in G_1$ but $(u, v, r)$ is not in $G_2$ where $u, v$ are vertices and $r$ is the relation. Because of this, suppose $w$ is a node with feature $r$ connected that is linked to both $u, v$ in the argument graph for both $G_1' = G_2'$. Namely both $(u, w)$ and $(w, v)$ are in $G_1' = G_2'$. Because the $(u, v, r)$ is not in $G_2$, $(w, v)$ is not constructed by the edge $(u, v, r)$, thus it must be constructed by another edge $(u', v, r)$. This suggests $w$ is connected with at least three vertices $u, v$ and $u'$. This is contradictory to the definition of the non-relational argument proxy graph. □

**Proof of Theorem 1**

*Proof.* Based on Lemma 1, as the baseline models perform message passing on the proxy graph, their expressiveness is as powerful as the 1-WL graph isomorphism test (Xu et al., 2019). Additionally, according to Lemma 1, the encoding process of LSGT on the session query graph is equivalent to using order-2 TokenGT on the proxy graph. Order-2 TokenGT can approximate the 2-IGN network (Kim et al., 2022), and the 2-IGN network is at least as expressive as the 2-WL graph isomorphism test (Maron et al., 2019). Since the 2-WL test is equivalent to the 1-WL test, we can conclude that LSGT has at least the same expressiveness as the baseline models.

□

**Proof of Theorem 2**

*Proof.* To prove the expressiveness of LSGT on the multi-relational proxy graph is at least 1-RWL, we need to show that for two non-isomorphic multi-relational graphs $G$ and $H$, if they can be distinguished by 1-RWL or equivalently CompGCN, then it also can be distinguished by LSGT. According to the CompGCN definition and the definition of the Non-Relational Argument Proxy Graph of $G'$ and $H'$ which are constructed from $G$ and $H$ respectively, CompGCN computed on $G$ and $H$ can be regarded as a message passing on the non-relational message passing on $G'$ and $H'$. Thus, if $G$ and $H$ can be distinguished by CompGCN, then $G'$ and $H'$ can be distinguished by a certain non-relational message-passing algorithm. Thus $G'$ and $H'$ can be distinguished by the 1-WL test. As Shown in previous proof LSGT is at least as powerful as the 2-WL test, and 1-WL and 2-WL tests are equivalent. We can conclude that LSGT is able to distinguish $G'$ and $H'$. According to Lemma 3, if LSGT is able to distinguish $G'$ and $H'$ then it is able to distinguish $G$ and $H$.

□

**Proof of Theorem 3**

*Proof.* Operation-wise permutation invariance mainly focuses on the `Intersection` and `Union` operations. Suppose the input vertices for such an operator are $\{p_1, p_2, \ldots, p_n\}$. If an arbitrary permutation over these vertices is denoted as $\{p'_1, p'_2, \ldots, p'_n\}$, a global permutation of token identifiers can be constructed, where vertices $p_i$ are mapped to $p'_i$ and the rest are mapped to themselves. As per Lemma 2, LSGT can approximate 2-IGN (Maron et al., 2019), which is permutation invariant. Therefore, LSGT can approximate a query encoder that achieves operation-wise permutation invariance. □

## B  THE CONCRETE MEANINGS OF VARIOUS QUERY TYPES

In this session, we describe concrete meanings of the query types shown in Figure 5. The meanings are listed in the Table 4.

## C  ABLATION STUDY

The ablation study is given in Table 6. In the first ablation study, we removed the tokens representing the logical structures. In the second ablation study, we removed the order information in the hypergraph by removing the positional encoding features of item tokens in each session.

When we removed the logical structure information, the model's performance drastically dropped, especially for queries involving negations and multi-hop reasoning, such as ip, pi, inp, and pin. Without the logical structure, the model could only use co-occurrence information like "bag of sessions" and "bag of items" to rank candidate answers. While this information may be useful for simple structured queries, it is not very useful for complex structured queries.

Similarly, when we removed the order information within each session, the overall performance also drastically dropped. This demonstrates two things: First, the item orders in each session are critical in this task. Second, the LSGT model is effectively able to utilize the order information for this task.

Table 4: The query types and their corresponding explanations.

| Query Types | Explanations |
|---|---|
| 1p | Predict the product that is desired by a given session. |
| 2p | Predict the attribute value of the product that is desired by a given session. |
| 2iA | Predict the product that is desired by a given session with a certain attribute value. |
| 2iS | Predict the product that is desired by both given sessions. |
| 3i | Predict the product that is desired by both given sessions with a certain attribute value. |
| ip | Predict the attribute value of the product that is desired by both of the given sessions. |
| pi | Predict the attribute value of the product that is desired by a given session, this attribute value is possessed by another given item. |
| 2u | Predict the product that is desired by either one of the sessions. |
| up | Predict the attribute value of the product that is desired by either of the sessions. |
| 2inA | Predict the product that is desired by a given session, but does not have a certain attribute. |
| 2inS | Predict the product that is desired by a given session, but is not wanted by another session. |
| 3in | Predict the product that is desired by a given session with a certain attribute, but is not wanted by another session. |
| inp | Predict the attribute value of the product that is desired by a given session, but is not wanted by another session. |
| pin | Predict the attribute value of the product that is desired by a given session, but is not possessed by another given item. |

Table 5: The query structures are used for training and evaluation. For brevity, the $p$, $i$, $n$, and $u$ represent the projection, intersection, negation, and union operations. The query types are trained and evaluated under supervised settings.

| | Train Queries | | Validation Queries | Test Queries |
|---|---|---|---|---|
| Dataset | Item-Attribute | Others | All Types | All Types |
| Amazon | 2,535,506 | 720,816 | 36,041 | 36,041 |
| Diginetica | 249,562 | 60,235 | 3,012 | 3,012 |
| Dressipi | 414,083 | 668,650 | 33,433 | 33,433 |

## D  FURTHER EVALUATION ON COMPOSITIONAL GENERALIZATION

We further conduct experiments on compositional generalization and the results are shown in Table 7. The newly included query types are 3iA, 3ip, 3inA, and 3inp. We selected these query types because they are complex, involve three anchors, cover both EPFO queries and queries involving negations, and have both 1-hop and 2-hop relational projections in the reasoning process. These query types were not trained during the training process but were evaluated in a zero-shot manner.

We compared their performance against the baselines and found that our proposed method showed stronger compositional generalization on these unseen query types. It achieved MRR improvement ranging from 1.28 to 3.22 on three datasets.

Table 6: The ablation study on the logical structure and item orders in each session.

| Dataset | Encoder | Average | 1p | 2p | 2ia | 2is | 3i | pi | ip | 2u | up | 2ina | 2ins | 3in | inp | pin |
|---|---|---|---|---|---|---|---|---|---|---|---|---|---|---|---|---|
| Amazon | LSGT | 31.99 | 17.73 | 9.10 | 56.73 | 84.26 | 69.39 | 19.39 | 19.47 | 15.40 | 7.86 | 20.98 | 22.00 | 60.70 | 35.95 | 8.84 |
| | w/o Logic Structure | 15.98 | 5.41 | 2.31 | 30.31 | 50.21 | 45.21 | 3.75 | 5.49 | 4.88 | 2.56 | 16.32 | 15.77 | 38.19 | 2.13 | 1.11 |
| | w/o Session Order | 8.45 | 6.29 | 2.59 | 17.22 | 13.85 | 19.34 | 14.07 | 3.23 | 3.49 | 1.73 | 5.50 | 4.75 | 17.54 | 4.92 | 3.73 |
| Diginetica | LSGT | 40.59 | 32.00 | 15.27 | 83.34 | 90.61 | 86.05 | 15.62 | 33.80 | 26.34 | 14.45 | 24.15 | 28.69 | 83.04 | 19.21 | 15.62 |
| | w/o Logic Structure | 27.17 | 18.61 | 3.84 | 68.40 | 62.80 | 64.87 | 10.13 | 20.22 | 16.08 | 8.49 | 17.38 | 14.17 | 60.37 | 9.21 | 5.74 |
| | w/o Session Order | 17.07 | 5.08 | 9.71 | 45.49 | 34.42 | 43.23 | 9.69 | 21.71 | 3.66 | 7.92 | 4.39 | 2.56 | 35.80 | 9.98 | 5.36 |
| Dressipi | LSGT | 70.22 | 31.12 | 96.16 | 64.26 | 76.85 | 78.66 | 98.02 | 96.98 | 28.83 | 96.04 | 25.58 | 30.66 | 65.93 | 97.74 | 96.30 |
| | w/o Logic Structure | 25.13 | 14.87 | 2.45 | 42.03 | 59.63 | 67.62 | 9.27 | 17.71 | 18.05 | 7.64 | 19.62 | 24.67 | 59.01 | 1.95 | 7.29 |
| | w/o Session Order | 39.78 | 9.21 | 42.80 | 21.57 | 19.57 | 23.28 | 88.31 | 61.47 | 6.31 | 68.27 | 7.41 | 6.87 | 15.96 | 96.53 | 89.35 |

Table 7: The out-of-distribution query types evaluation. We further evaluate four types of queries with types that are unseen during the training process.

| Dataset | Query Encoder | 3iA | 3ip | 3inA | 3inp | Average OOD |
|---|---|---|---|---|---|---|
| Amazon | FuzzQE + Attn-Mixer | 66.72 | 29.67 | 54.33 | 48.76 | 49.87 |
| | Q2P + Attn-Mixer | 33.51 | 11.42 | 51.47 | 41.46 | 34.47 |
| | NQE | 61.72 | 1.98 | 46.47 | 34,04 | 36.72 |
| | SQE + Transformers | 66.03 | 28.41 | 55.61 | 51.28 | 50.33 |
| | LSGT (Ours) | **68.44** | **34.22** | **58.50** | **51.49** | **53.16** |
| Diginetica | FuzzQE + Attn-Mixer | 88.30 | 32.88 | 82.75 | 34.50 | 59.61 |
| | Q2P + Attn-Mixer | 40.28 | **43.93** | 54.31 | **48.20** | 46.68 |
| | NQE | 86.25 | 20.79 | 64.74 | 20.93 | 48.18 |
| | SQE + Transformers | 88.05 | 31.33 | 81.77 | 35.83 | 59.25 |
| | LSGT (Ours) | **91.71** | 35.24 | **83.30** | 41.05 | **62.83** |
| Dressipi | FuzzQE + Attn-Mixer | 65.43 | 95.64 | 53.36 | 97.75 | 78.05 |
| | Q2P + Attn-Mixer | 60.64 | 96.78 | 52.22 | 97.28 | 76.73 |
| | NQE | 31.96 | 96.18 | 9.89 | 97.80 | 58.96 |
| | SQE + Transformers | 72.61 | 97.12 | 55.20 | 98.14 | 80.77 |
| | LSGT (Ours) | **74.34** | **97.30** | **58.30** | **98.23** | **82.04** |

