# OpenReview forum: "Understanding Inter-Session Intentions via Complex Logical Reasoning"
_ICLR.cc/2024/Conference — Submitted to ICLR 2024_

### Official Review · Reviewer_kouL · 2023-10-29

**Soundness:** 3 good
**Presentation:** 3 good
**Contribution:** 3 good
**Rating:** 6
**Confidence:** 4

**Summary:**

This paper introduces the task of Logical Session Query Answering (LSQA) and presents a solution called the Logical Session Graph Transformer (LSGT) model. The objective of LSQA is to learn logical queries for observed user interaction sessions. This task could help understand the logical intention of user interactions. The LSGT model achieves this by uniformly representing sessions, items, relations, and logical operators as tokens and leveraging a transformer-based sequential model for encoding.

The paper provides a theoretical analysis that primarily focuses on demonstrating the expressiveness of the proposed LSGT model. Additionally, comprehensive experiments are conducted to validate the superiority of the proposed model compared to existing baselines.

**Strengths:**

- This paper proposes the task of Logical Session Query Answering (LSQA), providing an novel paradigm for enhancing applications like session-based recommendation and query recommendation by understanding the logical structures of users' latent intents.
- The paper provides a theoretical analysis on the expressiveness of the proposed Logical Session Graph Transformer (LSGT) model.
- The paper innovatively build a unified representation model for items, sessions and logical operators using hypergraphs and sequential models.

**Weaknesses:**

- Though the proposed task is novel, the proposed technical solution LSGT relies on existing hypergraph structures and transformer architeactures. Such designs have limited differences compared to existing sequential models and graph models. This lower the technical contribution of this paper.
- The evaluation part could be enhanced with more diverse experiments to conduct a more comprehensive empirical study, such as ablation study, hyperparameter study, case study on the generated queries, and an investigations on the benefits of LSGT brought to downstream tasks like session-based recommendation.

Minor mistake: In the summary for contributions: "We propose to propose ..."

**Questions:**

My concerns would be alleviated if the authors could provide further clarification on the technical novelty aspect and the comprehensiveness of the experiments. Please refer to the weaknesses part for details.

---

> ### Author Response · Authors · 2023-11-22
>
> Re W1
>
> As part of our technical contribution, LSGT proposes a new strategy of using tokens to represent sessions and logical structures. This approach demonstrates its novelty and advantage by achieving an average improvement of 2.32 on complex structures including negations, such as 2ina, 2ins, 3in, inp, and pin, over three datasets compared to a model that also uses a Transformer and backbone (SQE).
>
> Furthermore, we conducted additional experiments on four previously unseen query types (3iA, 3ip, 3inA, and 3inp) and showed that our method achieves an average improvement of 2.44 in MRR over three datasets compared to a previous method that also uses a Transformer encoder (SQE+Transformer). The details of this experiment are provided in Re W2.
>
>
>
>
> Re W2
>
> Following your suggestions, we included two ablation studies on the hypergraph and logical structures. In the first study, we removed the tokens representing the logical structures. In the second study, we removed the order information in the hypergraph by removing the positional encoding features of item tokens in each session.
>
> Here are the results:
>
> | Dataset    | Encoder             | Average |   1p  |   2p  |  2ia  |  2is  |   3i  |   pi  |   ip  |   2u  |   up  | 2ina  | 2ins  | 3in   | inp   | pin   |
> |------------|---------------------|---------|:-----:|:-----:|:-----:|:-----:|:-----:|:-----:|:-----:|:-----:|:-----:|-------|-------|-------|-------|-------|
> | Amazon     | LSGT                |  31.99  | 17.73 |  9.10 | 56.73 | 84.26 | 69.39 | 19.39 | 19.47 | 15.40 |  7.86 | 20.98 | 22.00 | 60.70 | 35.95 |  8.84 |
> |            | w/o Logic Structure |  15.98  |  5.41 |  2.31 | 30.31 | 50.21 | 45.21 |  3.75 |  5.49 |  4.88 |  2.56 | 16.32 | 15.77 | 38.19 |  2.13 |  1.11 |
> |            | w/o Session Order   |   8.45  |  6.29 |  2.59 | 17.22 | 13.85 | 19.34 | 14.07 |  3.23 |  3.49 |  1.73 |  5.50 |  4.75 | 17.54 |  4.92 |  3.73 |
> | Diginetica | LSGT                |  40.59  | 32.00 | 15.27 | 83.34 | 90.61 | 86.05 | 15.62 | 33.80 | 26.34 | 14.45 | 24.15 | 28.69 | 83.04 | 19.21 | 15.62 |
> |            | w/o Logic Structure |  27.17  | 18.61 |  3.84 | 68.40 | 62.80 | 64.87 | 10.13 | 20.22 | 16.08 |  8.49 | 17.38 | 14.17 | 60.37 |  9.21 |  5.74 |
> |            | w/o Session Order   |  17.07  |  5.08 |  9.71 | 45.49 | 34.42 | 43.23 |  9.69 | 21.71 |  3.66 |  7.92 |  4.39 |  2.56 | 35.80 |  9.98 |  5.36 |
> | Dressipi   | LSGT                |  70.22  | 31.12 | 96.16 | 64.26 | 76.85 | 78.66 | 98.02 | 96.98 | 28.83 | 96.04 | 25.58 | 30.66 | 65.93 | 97.74 | 96.30 |
> |            | w/o Logic Structure |  25.13  | 14.87 |  2.45 | 42.03 | 59.63 | 67.62 |  9.27 | 17.71 | 18.05 |  7.64 | 19.62 | 24.67 | 59.01 |  1.95 |  7.29 |
> |            | w/o Session Order   |  39.78  |  9.21 | 42.80 | 21.57 | 19.57 | 23.28 | 88.31 | 61.47 |  6.31 | 68.27 |  7.41 |  6.87 | 15.96 | 96.53 | 89.35 |
>
> Removing the logical structure information drastically dropped the model's performance, especially for queries involving negations and multi-hop reasoning, such as ip, pi, inp, and pin. Without the logical structure, the model could only use co-occurrence information like "bag of sessions" and "bag of items" to rank candidate answers. While this information may be useful for simple structured queries, it is not very useful for complex structured queries.
>
> Similarly, removing the order information within each session also drastically dropped the overall performance. This demonstrates two things: First, the item orders in each session are critical in this task. Second, the LSGT model is effectively able to utilize the order information for this task.
>
> In addition, we included a new evaluation on the compositional generalization of LSGT. The newly included query types are 3iA, 3ip, 3inA, and 3inp. We selected them because they are all complex types involving three anchors, cover both EPFO queries and queries involving negations, and have both 1-hop and 2-hop relational projections in the reasoning process. These query types were not trained during the training process but were evaluated in a zero-shot manner.
>
> Here are their performances against the baselines:

---

> > ### Author Response · Authors · 2023-11-22
> > **Continued**
> >
> > | Dataset | Query Encoder | 3iA | 3ip | 3inA | 3inp | Average OOD |
> > |---|---|:---:|:---:|:---:|:---:|:---:|
> > | Amazon | FuzzQE + Attn-Mixer | 66.72 | 29.67 | 54.33 | 48.76 | 49.87 |
> > |  | Q2P + Attn-Mixer | 33.51 | 11.42 | 51.47 | 41.46 | 34.47 |
> > |  | NQE | 61.72 | 1.98 | 46.47 | 34,04 | 36.72 |
> > |  | SQE + Transformers | 66.03 | 28.41 | 55.61 | 51.28 | 50.33 |
> > |  | LSGT (Ours) | **68.44** | **34.22** | **58.50** | **51.49** | **53.16 (+2.83)** |
> > | Diginetica | FuzzQE + Attn-Mixer | 88.30 | 32.88 | 82.75 | 34.50 | 59.61 |
> > |  | Q2P + Attn-Mixer | 40.28 | **43.93** | 54.31 | **48.20** | 46.68 |
> > |  | NQE | 86.25 | 20.79 | 64.74 | 20.93 | 48.18 |
> > |  | SQE + Transformers | 88.05 | 31.33 | 81.77 | 35.83 | 59.25 |
> > |  | LSGT (Ours) | **91.71** | 35.24 | **83.30** | 41.05 | **62.83 (+3.22)** |
> > | Dressipi | FuzzQE + Attn-Mixer | 65.43 | 95.64 | 53.36 | 97.75 | 78.05 |
> > |  | Q2P + Attn-Mixer | 60.64 | 96.78 | 52.22 | 97.28 | 76.73 |
> > |  | NQE | 31.96 | 96.18 | 9.89 | 97.80 | 58.96 |
> > |  | SQE + Transformers | 72.61 | 97.12 | 55.20 | 98.14 | 80.77 |
> > |  | LSGT (Ours) | **74.34** | **97.30** | **58.30** | **98.23** | **82.04 (+1.28)** |
> >
> > Our proposed method demonstrates stronger compositional generalization on these previously unseen query types and achieves MRR improvements ranging from 1.28 to 3.22 on three datasets.

---

### Official Review · Reviewer_Kyoc · 2023-10-29

**Soundness:** 2 fair
**Presentation:** 3 good
**Contribution:** 2 fair
**Rating:** 5
**Confidence:** 5

**Summary:**

The paper formulates an item recommendation task based on the previous session history as a complex logical graph query (named as Logical Session Query Answering). In such a query, items and attributes are nodes, several items can be connected in sessions (hyperedges denoting the order of obtaining the items), relations form projection operators in a query, and other logical operators (intersection, union, negation) combine nodes and projections into a single complex query. Instead of operating on the complete hypergraph of items, sessions, and attributes, the authors decide to operate on the single query level and predict answer entities directly after linearizing the query via the Logical Session Graph Transformer (essentially, a TokenGT from [1]). The authors prove that their transformer is permutation invariant (with respect to intersection and union operators), and run experiments on 3 datasets showing marginal improvements over the baselines.

**Strengths:**

**S1.** The item recommendation task is framed as a complex logical query. While the task per se is not new (LogiRec [2] originally introduced it with projection and intersection operators), this paper extends it to unions and negations and to hypergraphs.

**S2.** Evaluation includes several baselines (that show, on the other hand, that the proposed approach only marginally outperforms existing models, but more on that in W2)

**Weaknesses:**

Starting from the claimed contributions:

**W1. Task.** The formulated task of Logical Session Query Answering is essentially query answering over hypergraphs. Sessions are n-ary edges, and other relations form 2-ary edges, so the hypergraph has edges of different arity. The temporal aspect of items in session hyperedges (that items follow each other in one session) seems to be of little use as the best-performing models are not using this information anyway. I would recommend the authors to focus the contribution on extending complex query answering to hypergraphs as there is not that much work in that subfield (StarQE is for hyper-relational graphs, and NQE supports both hyper-relational and hypergraphs).

**W2. Encoder + Experimental results.** The proposed logical session graph transformer (LSGT) is just one of the many query linearization strategies, eg, BiQE [3], kgTransformer [4], or SQE [5] that convert the query graph into a sequence with some positional information to be sent jointly into a Transformer. Architecture-wise, LSGT is TokenGT [1] but with a slightly different input format that sends tokens of logical operators. Experimentally, LSGT is very close to SQE [5] (the gap is often <1 MRR point) so it is hard to claim any novelty or effectiveness in this linearization strategy or in a slightly different transformer encoder.

**W3. Theory.** The theoretical study in Section 4.5 is derived from TokenGT and seems to be hardly applicable to the case of logical query answering. TokenGT’s theory of WL expressiveness assumes the graphs are non-relational whereas all logical query graphs studied in this work are relational, i.e., they have labeled edge types. There is a different line of work studying expressiveness of GNNs over relational graphs [6,7] and I would recommend starting from them in order to derive any expressiveness claims. Permutation invariance proofs are rather trivial because the Transformer architecture itself is permutation equivariant.

Overall, I think the paper has more potential if:
* The authors frame the task as the hypergraph query answering with the full support of first-order logical operators (intersection, union, negation) and demonstrate that several existing Transformer-based models show similar results on 3 benchmarks despite different linearization strategies;
* Tone down the claims on the _logical session_ QA (it’s a hypergraph), new graph transformer and its expressiveness (TokenGT is not new, theory for non-relational graphs does not apply to relational ones), and state-of-the-art (all Transformer-based models show a very similar performance).

I understand that it would require substantial re-writing of several sections, so I am willing to increase the score if the authors decide to do it during the discussion period.

Minor comments:
* Too many sentences (especially in Section 3) start with noisy and artificial “however” and “meanwhile”. You don’t have to contrast every sentence to each other every time.
* $p$ and $q$ denote different things in 4.2 (item and session) and 4.3 (just two nodes) and it is confusing.
* 4.4 Learning LSGT -> Training LSGT

**References**

[1] Kim et al. Pure transformers are powerful graph learners. NeurIPS 2022.
[2] Tang et al. LogicRec: Recommendation with Users' Logical Requirements. SIGIR’23.
[3] Kotnis et al. Answering complex queries in knowledge graphs with bidirectional sequence encoders. AAAI 2021.
[4] Liu et al. Mask and reason: Pre-training knowledge graph transformers for complex logical queries. KDD’22.
[5] Bai et al. Sequential query encoding for complex query answering on knowledge graphs. TMLR 2023.
[6] Barcelo et al. Weisfeiler and Leman Go Relational. LOG 2022.
[7] Huang et al. A theory of link prediction via relational Weisfeiler-Leman. NeurIPS 2023.

**Questions:**

N/A

---

> ### Author Response · Authors · 2023-11-22
>
> Thank you for providing such constructive feedback. We have put in significant effort to revise the paper based on your comments and suggestions, and all the rewritten parts are highlighted in blue.
>
> Re W1:
>
> Regarding the task formulation, we have taken your advice and extended the problem definition of CQA to include sessions as ordered hypergraphs of varying arity, namely LS-CQA. We also want to emphasize that the order of items in the session is crucial for recommending relevant items, and top-performing models such as SQE, LSGT, and session encoders all take this into account.
>
> Furthermore, we want to highlight the novelty of our problem formulation, as we are the first to bridge the CQA problem with session-based recommendations. CQA methods and knowledge graph reasoning are essential for session understanding, and our approach is unique in that it addresses user anonymity, which is a common feature of session-based recommendation tasks. By incorporating logical graph reasoning methods, we believe our approach is not only novel but also useful for improving session-based recommendation systems.
>
>
> Re W2 Encoder + Experiments:
>
> We have taken your advice and demonstrated that transformer-based methods have comparable performance despite using different linearization strategies. However, we want to highlight that the linearization strategy employed in LSGT outperforms others in queries with negations.
>
> Furthermore, we have conducted additional experiments to investigate the compositional generalization of different linearization strategies. Our results show that the linearization strategy used in LSGT exhibits better compositional generalization than other existing strategies. Below are the detailed performance results:
>
>
> | Dataset | Query Encoder | 3iA | 3ip | 3inA | 3inp | Average OOD |
> |---|---|:---:|:---:|:---:|:---:|:---:|
> | Amazon | FuzzQE + Attn-Mixer | 66.72 | 29.67 | 54.33 | 48.76 | 49.87 |
> |  | Q2P + Attn-Mixer | 33.51 | 11.42 | 51.47 | 41.46 | 34.47 |
> |  | NQE | 61.72 | 1.98 | 46.47 | 34,04 | 36.72 |
> |  | SQE + Transformers | 66.03 | 28.41 | 55.61 | 51.28 | 50.33 |
> |  | LSGT (Ours) | **68.44** | **34.22** | **58.50** | **51.49** | **53.16 (+2.83)** |
> | Diginetica | FuzzQE + Attn-Mixer | 88.30 | 32.88 | 82.75 | 34.50 | 59.61 |
> |  | Q2P + Attn-Mixer | 40.28 | **43.93** | 54.31 | **48.20** | 46.68 |
> |  | NQE | 86.25 | 20.79 | 64.74 | 20.93 | 48.18 |
> |  | SQE + Transformers | 88.05 | 31.33 | 81.77 | 35.83 | 59.25 |
> |  | LSGT (Ours) | **91.71** | 35.24 | **83.30** | 41.05 | **62.83 (+3.22)** |
> | Dressipi | FuzzQE + Attn-Mixer | 65.43 | 95.64 | 53.36 | 97.75 | 78.05 |
> |  | Q2P + Attn-Mixer | 60.64 | 96.78 | 52.22 | 97.28 | 76.73 |
> |  | NQE | 31.96 | 96.18 | 9.89 | 97.80 | 58.96 |
> |  | SQE + Transformers | 72.61 | 97.12 | 55.20 | 98.14 | 80.77 |
> |  | LSGT (Ours) | **74.34** | **97.30** | **58.30** | **98.23** | **82.04 (+1.28)** |
>
>
>
> Re W3 Theory:
>
> We have read and analyzed the reference papers [1,2], and we developed a new expressive claim based on the relational expressiveness of LSGT. We proved that the LSGT is at least as powerful as 1-RWL, namely the expressiveness of R-GCN and CompGCN. We include the proof in the appendix and mark it in blue.
>
>
> [1] Barcelo et al. Weisfeiler and Leman Go Relational. LOG 2022.
> [2] Huang et al. A theory of link prediction via relational Weisfeiler-Leman. NeurIPS 2023.
>
> We really appreciate your constructive feedback. If you have any other concerns please do not hesitate to tell us.

---

### Official Review · Reviewer_kt6F · 2023-10-31

**Soundness:** 3 good
**Presentation:** 2 fair
**Contribution:** 3 good
**Rating:** 6
**Confidence:** 3

**Summary:**

In this work, the authors focus on product and attribute recommendation by modeling complex user intention. They employ the logical session query answering (LSQA) to formulate the task. The proposed logical session graph transformer (LSGT) model runs on a hyper session graph, which uses a standard transformer structure to encode different entities. Experiments on three real-world datasets demonstrate the effectiveness of LSGT for complex session query answering.

**Strengths:**

1. The motivation that incorporates logical session query answering into product recommendation to model user intent is novel.
2. The experimental results demonstrate the effectiveness of the proposed LSGT.
3. The authors theoretically justify the expressiveness and operator-wise permutation invariance of LSGT.

**Weaknesses:**

1. There are some obvious typos. Authors should scrutinize the writing of the paper.
(1) In the 5th line of section 4.3, the formula after “The edge feature is denoted as” lacks a proper superscript.
(2) In Table 5, the first word “Predicti” in explanation of query type 2p should be “Predict”.
(3) In Table 5, the word “prodict” in explanation of query type ip should be “product”.
(4) In the 2nd line below Figure 5, the word “descibed” should be “described”.
2. In Figure 5, the query structure of ip is the same as up and the query structure of 2iS is the same as 2uS. It would be better to distinguish them like [1].
3. The paper lacks detailed description for figures especially Figure 3, which is hard to understand for readers.
4. It would be better to evaluate the model’s generalization ability of unseen query structures like [1,2,3].

[1] Jiaxin Bai, Zihao Wang, Hongming Zhang, and Yangqiu Song. 2022. Query2Particles: Knowledge Graph Reasoning with Particle Embeddings. In Findings of the Association for Computational Linguistics: NAACL 2022, pages 2703–2714, Seattle, United States. Association for Computational Linguistics.
[2] Chen, X., Hu, Z., & Sun, Y. (2022). Fuzzy Logic Based Logical Query Answering on Knowledge Graphs. Proceedings of the AAAI Conference on Artificial Intelligence, 36(4), 3939-3948.
[3] Jiaxin Bai, Tianshi Zheng, and Yangqiu Song. Sequential query encoding for complex query answering on knowledge graphs. Transactions on Machine Learning Research, 2023. ISSN 2835-8856

**Questions:**

1. Why do authors not evaluate the model’s generalization ability of unseen query structures like existing works?
2. Is there an explanation for the author's choice of 14 query structures? Can some other query structures like 2i, and pni be incorporated?
3. Is it possible to make an ablation study for hypergraph and logical reasoning?

---

> ### Author Response · Authors · 2023-11-22
> **Review Discussions**
>
> Thank you for your review, we would like to address your concerns individually.
>
> Re W1
>
> Thank you so much, we have fixed the typos you mentioned and marked them in blue in the revised paper. We have also conducted a thorough check on the grammatical issues and typos and also fixed them at the same time.
>
> Re W2
>
> We have made some updates to our work. Specifically, we added the notation of "u" and "n" as in previous work to distinguish between “2iS” and “2uS”. Additionally, we included query structures for the four out-of-distribution query types in Figure 5, which addresses your concerns on compositional generalizability. You can find more details in Re W4.
>
> Re W3
>
> We've made some updates to our paper. Specifically, we added a detailed description for Figure 3 to better explain our problem setting and relevant query encoding methods. We marked the newly added explanations in blue for your convenience.
>
> Re W4
>
> We have followed your suggestions and added new experiment results on compositional generalization. The newly included query types are 3iA, 3ip, 3inA, and 3inp. We selected these query types because they are complex, involve three anchors, cover both EPFO queries and queries involving negations, and have both 1-hop and 2-hop relational projections in the reasoning process. These query types were not trained during the training process but were evaluated in a zero-shot manner.
>
>
> | Dataset | Query Encoder | 3iA | 3ip | 3inA | 3inp | Average OOD |
> |---|---|:---:|:---:|:---:|:---:|:---:|
> | Amazon | FuzzQE + Attn-Mixer | 66.72 | 29.67 | 54.33 | 48.76 | 49.87 |
> |  | Q2P + Attn-Mixer | 33.51 | 11.42 | 51.47 | 41.46 | 34.47 |
> |  | NQE | 61.72 | 1.98 | 46.47 | 34,04 | 36.72 |
> |  | SQE + Transformers | 66.03 | 28.41 | 55.61 | 51.28 | 50.33 |
> |  | LSGT (Ours) | **68.44** | **34.22** | **58.50** | **51.49** | **53.16 (+2.83)** |
> | Diginetica | FuzzQE + Attn-Mixer | 88.30 | 32.88 | 82.75 | 34.50 | 59.61 |
> |  | Q2P + Attn-Mixer | 40.28 | **43.93** | 54.31 | **48.20** | 46.68 |
> |  | NQE | 86.25 | 20.79 | 64.74 | 20.93 | 48.18 |
> |  | SQE + Transformers | 88.05 | 31.33 | 81.77 | 35.83 | 59.25 |
> |  | LSGT (Ours) | **91.71** | 35.24 | **83.30** | 41.05 | **62.83 (+3.22)** |
> | Dressipi | FuzzQE + Attn-Mixer | 65.43 | 95.64 | 53.36 | 97.75 | 78.05 |
> |  | Q2P + Attn-Mixer | 60.64 | 96.78 | 52.22 | 97.28 | 76.73 |
> |  | NQE | 31.96 | 96.18 | 9.89 | 97.80 | 58.96 |
> |  | SQE + Transformers | 72.61 | 97.12 | 55.20 | 98.14 | 80.77 |
> |  | LSGT (Ours) | **74.34** | **97.30** | **58.30** | **98.23** | **82.04 (+1.28)** |
>
> We compared their performance against the baselines and found that our proposed method showed stronger compositional generalization on these unseen query types. It achieved MRR improvement ranging from 1.28 to 3.22 on three datasets.
>
> Re Q1
> We have added the evaluation of compositional generalization, as in previous work, and included the results in the original paper. Initially, we did not include compositional generalization because we believed it was not very important in the real setting of session reasoning. We thought that if we wanted to improve performance on some unseen query types, a better approach would be to sample more data with that query type and conduct supervised training on it.
>
> However, we now realize the importance of compositional generalization in the general problem of complex query answering. Therefore, we further included the experiment in Re W4 to demonstrate the effectiveness of our method on compositional generalization.
>
>
> Re Q2
>
> We selected and derived fourteen query types from the original fourteen types of queries in [1,2]. The 2iS and 2iA are both derived from 2i queries. They involve 1 session 1 item and 2 sessions as their anchors, respectively.
>
> Regarding pni, we found it difficult to interpret its practical meaning in a real problem setting. It can be interpreted as "Predict the attribute value of a given product that is not desired by a given session." However, we believe that predicting the attribute value for a single item is not meaningful in session understanding. Therefore, we did not include this type in our training and evaluation so that we could focus on other important query types.

---

> > ### Author Response · Authors · 2023-11-22
> > **Continued**
> >
> > Re Q3
> >
> > We followed your suggestions and included two ablation studies on the hypergraph and logical structures, respectively. In the first ablation study, we removed the tokens representing the logical structures. In the second ablation study, we removed the order information in the hypergraph by removing the positional encoding features of item tokens in each session.
> >
> > The results are as follows:
> >
> > | Dataset    | Encoder             | Average |   1p  |   2p  |  2ia  |  2is  |   3i  |   pi  |   ip  |   2u  |   up  | 2ina  | 2ins  | 3in   | inp   | pin   |
> > |------------|---------------------|---------|:-----:|:-----:|:-----:|:-----:|:-----:|:-----:|:-----:|:-----:|:-----:|-------|-------|-------|-------|-------|
> > | Amazon     | LSGT                |  31.99  | 17.73 |  9.10 | 56.73 | 84.26 | 69.39 | 19.39 | 19.47 | 15.40 |  7.86 | 20.98 | 22.00 | 60.70 | 35.95 |  8.84 |
> > |            | w/o Logic Structure |  15.98  |  5.41 |  2.31 | 30.31 | 50.21 | 45.21 |  3.75 |  5.49 |  4.88 |  2.56 | 16.32 | 15.77 | 38.19 |  2.13 |  1.11 |
> > |            | w/o Session Order   |   8.45  |  6.29 |  2.59 | 17.22 | 13.85 | 19.34 | 14.07 |  3.23 |  3.49 |  1.73 |  5.50 |  4.75 | 17.54 |  4.92 |  3.73 |
> > | Diginetica | LSGT                |  40.59  | 32.00 | 15.27 | 83.34 | 90.61 | 86.05 | 15.62 | 33.80 | 26.34 | 14.45 | 24.15 | 28.69 | 83.04 | 19.21 | 15.62 |
> > |            | w/o Logic Structure |  27.17  | 18.61 |  3.84 | 68.40 | 62.80 | 64.87 | 10.13 | 20.22 | 16.08 |  8.49 | 17.38 | 14.17 | 60.37 |  9.21 |  5.74 |
> > |            | w/o Session Order   |  17.07  |  5.08 |  9.71 | 45.49 | 34.42 | 43.23 |  9.69 | 21.71 |  3.66 |  7.92 |  4.39 |  2.56 | 35.80 |  9.98 |  5.36 |
> > | Dressipi   | LSGT                |  70.22  | 31.12 | 96.16 | 64.26 | 76.85 | 78.66 | 98.02 | 96.98 | 28.83 | 96.04 | 25.58 | 30.66 | 65.93 | 97.74 | 96.30 |
> > |            | w/o Logic Structure |  25.13  | 14.87 |  2.45 | 42.03 | 59.63 | 67.62 |  9.27 | 17.71 | 18.05 |  7.64 | 19.62 | 24.67 | 59.01 |  1.95 |  7.29 |
> > |            | w/o Session Order   |  39.78  |  9.21 | 42.80 | 21.57 | 19.57 | 23.28 | 88.31 | 61.47 |  6.31 | 68.27 |  7.41 |  6.87 | 15.96 | 96.53 | 89.35 |
> >
> > When we removed the logical structure information, the model's performance drastically dropped, especially for queries involving negations and multi-hop reasoning, such as ip, pi, inp, and pin. Without the logical structure, the model could only use co-occurrence information like "bag of sessions" and "bag of items" to rank candidate answers. While this information may be useful for simple structured queries, it is not very useful for complex structured queries.
> >
> > Similarly, when we removed the order information within each session, the overall performance also drastically dropped. This demonstrates two things: First, the item orders in each session are critical in this task. Second, the LSGT model is effectively able to utilize the order information for this task.
> >
> > [1] Jiaxin Bai, Zihao Wang, Hongming Zhang, and Yangqiu Song. 2022. Query2Particles: Knowledge Graph Reasoning with Particle Embeddings. In Findings of the Association for Computational Linguistics: NAACL 2022, pages 2703–2714, Seattle, United States. Association for Computational Linguistics.
> > [2] Chen, X., Hu, Z., & Sun, Y. (2022). Fuzzy Logic Based Logical Query Answering on Knowledge Graphs. Proceedings of the AAAI Conference on Artificial Intelligence, 36(4), 3939-3948.
> > [3] Jiaxin Bai, Tianshi Zheng, and Yangqiu Song. Sequential query encoding for complex query answering on knowledge graphs. Transactions on Machine Learning Research, 2023. ISSN 2835-8856

---

### Meta-Review · Area_Chair_KPSW · 2023-12-06

**Metareview:**

This paper presents a novel method for understanding complex user intentions across multiple sessions.  The authors introduce the Logical Session Graph Transformer (LSGT) as an approach for encoding complex queries sessions as hypergraphs. They transform items, sessions, relation features, session structures, and logical structures into tokens, which are then encoded using a standard transformer model. This transformation enables them to effectively capture interactions among items in different sessions through the any-to-any attention mechanisms in transformer models.  In contrast to prior work where complex query answering is usually conducted on a knowledge graph G = (V, R), but this model also includes sessions and attribute values and their graph definition is therefore G = (V, R, S). where V is the set of vertices v, and the R is the set of relation r.  The proposed LSGT method outperforms all other models and is the current state-of-the-art for the task.

Strengths:
-The authors present a new way of encoding multi session user interactions that outperforms current state of the art methods.
-The motivation that incorporates logical session query answering into product recommendation to model user intent is novel.
-The authors theoretically justify the expressiveness and operator-wise permutation invariance of LSGT.

Weaknesses:

While the authors did substantial work to address the weaknesses outlined by the reviewers, including updating graphs, captions, references and adding new experiments, these significant additions failed to overcome the following weaknesses:

- Limited novelty - one reviewer expresses concern that tis is simply a re-labeling of existing tasks under with the addition of a linearization    strategy to represent things as a sequence of tokens for transformers;
- Improvements over the baselines was considered marginal
- Reviewers were not entirely convinces that the results of message passing over a relational graph were equivalent to message passing over an augmented graph which weakens the proof.

Overall this was considered a borderline paper, but one that needed revision.  With the additional work the authors have done for the rebuttal incorporated into a new submission, it will likely make a strong submission to a near future machine learning conference.

**Justification For Why Not Higher Score:**

The paper seems to have marginal novelty as highlighted by several of the reviewers.

**Justification For Why Not Lower Score:**

The method does seem to beat the state of the art and there seems to be no fatal flaw in it so I would prefer to see it accepted.

---

### Decision · Program_Chairs · 2024-01-16

Reject